



# Mid-Holocene rainfall changes in the southwestern Pacific

Cinthya Nava-Fernandez[1], Tobias Braun[2], Bethany Fox[3], Adam Hartland[4], Ola Kwiecien[5], Chelsea L. Pederson[1], Sebastian Hoepker[4], Stefano Bernasconi[6], Madalina Jaggi[6], John Hellstrom[7], Fernando Gázquez[8,9], Amanda French[4], Norbert Marwan[2], Adrian Immenhauser[1,10], Sebastian F.M. Breitenbach[5]

[1]Sediment- and Isotope Geology, Institute for Geology, Mineralogy and Geophysics, Ruhr-Universität Bochum, Universitätsstr. 150, 44801 Bochum, Germany
[2]Potsdam Institute for Climate Impact Research (PIK), Member of the Leibniz Association, Potsdam, Germany
[3]Department of Biological and Geographical Sciences, School of Applied Sciences, University of Huddersfield, Queensgate, Huddersfield, UK
[4]Environmental Research Institute, School of Science, Faculty of Science and Engineering, University of Waikato, Hamilton, Waikato, New Zealand
[5]Department of Geography and Environmental Sciences, Northumbria University, Newcastle upon Tyne, NE1 8ST, UK
[6]Department of Earth Sciences, ETH Zurich, Sonneggstrasse 5, 8092 Zurich, Switzerland
[7]School of Earth Sciences, The University of Melbourne, Australia
[8]Department of Biology and Geology, Universidad de Almería, Almería, 04120, Spain
[9]Andalusian Centre for the Monitoring and Assessment of Global Change (CAESCG), University of Almería, Spain
[10]Fraunhofer Research Institution for Energy Infrastructures and Geothermal Systems IEG, Am Hochschulcampus 1, 44801 Bochum, Germany

*Correspondence to*: Cinthya Nava-Fernandez (cinthya.navafernandez@rub.de)

**Abstract.**

A better understanding of ENSO dynamics is essential for modelling future climate change and its impacts on the ecosystems and lives of the inhabitants of the tropical Pacific islands, which face considerable environmental risk in the coming decades. This study reconstructs past ENSO dynamics using a multi-proxy approach applied to a stalagmite from Niue Island that covers the period 6.4-5.4 ka BP. $\delta^{18}O$ and $\delta^{13}C$, trace-element concentrations and image analysis are linked to an age-depth model constrained by eight U/Th dates and a complete lamina count.

Principal component analysis of the proxy time series reveals hydrological changes at seasonal scale that are expressed in differential stalagmite lamina growth and geochemical characteristics. Increased concentrations of host-rock derived elements (Mg/Ca and U/Ca) and higher $\delta^{18}O$ and $\delta^{13}C$ values are observed in the dark, dense calcite laminae that are deposited during the dry season, whereas during the wet season higher concentrations of soil derived elements (Zn/Ca, Mn/Ca) and higher $\delta^{18}O$ and $\delta^{13}C$ values are found in pale, porous calcite laminae. Greyscale intensity values measured along the stalagmite growth axis are used here as an indicator of colour and density changes of the alternating laminae, allowing for the construction of a further seasonality record which expresses the contrast between wet and dry seasons. The multi-proxy record from Niue shows seasonal cycles associated with hydrological changes controlled by the South Pacific convergence zone. Wavelet analysis of the greyscale record reveals that ENSO was continuously active during the depositional period, with two weaker intervals at 6-5.9 and 5.6-5.5 ka BP. ENSO activity is also observed in the seasonality record, but muted periods are more prolonged, and intervals of significant ENSO-band power are more episodic. Recurrence



analysis of nonlinear behaviour shows the influence that ENSO activity exerts on seasonality patterns and allows us to quantify the predictability of the climate system. Our results suggest that recurrence in the seasonal cycle of rainfall was reduced during periods when ENSO activity was stronger, pushing the system towards stochastic conditions. The tipping points from stochastic to predictable conditions may represent transitions in the Tropical Pacific mean state.

## 1. Introduction

Tropical Pacific dynamics play a key role in global climate. The South Pacific Convergence Zone (SPCZ) is the major climate feature that channels convective rainfall in the south Pacific at a seasonal scale (Brown et al., 2020). At interannual scales, the coupled ocean-atmosphere phenomenon El Niño-Southern Oscillation (ENSO) controls climate variability (Timmermann et al., 2018). These climate features modulate rainfall amount as well as the frequency and intensity of extreme climate events (e.g., tropical cyclones and droughts), and therefore these modes of climate variability greatly impact the ecosystems and the lives of the inhabitants of Pacific islands and the east and west coasts of the Pacific basin (Meteorology and CSIRO, 2011). A range of projections resulting from climate models have forecast the high vulnerability of the tropical Pacific islands to the impacts of anthropogenically driven warming (Meteorology and CSIRO, 2011). Accurately dated, high-resolution and long-term records of climate and ENSO variability are essential for evaluating the robustness of climate models and of forecasts of regional and global climate changes (Cane, 2005; Capotondi et al., 2015; Emile-Geay et al., 2016).

Diverse studies across the Pacific have reconstructed ENSO during the Holocene using climatic archives such as marine sediments (Koutavas et al., 2006), corals (Cobb et al., 2013; Tudhope et al., 2001), clastic lake sediments (Conroy et al., 2008; Moy et al., 2002), and speleothems (Chen et al., 2016). The resolution of these records ranges from multi-decadal to monthly, but they often represent discrete points in time rather than continuous intervals. Several studies from the Pacific have concluded that ENSO variability was reduced in terms of intensity and frequency during the mid-Holocene. These include eastern Pacific records such as lake sediments from Ecuador and the Galapagos (Conroy et al., 2008; Rodbell et al., 1999) and foraminifera from deep-sea sediments (Koutavas and Joanides, 2012), as well as western Pacific records such as pollen from Australasia (Shulmeister and Lees, 1995) and corals from Papua New Guinea (Tudhope et al., 2001). According to a numerical model (Clement et al., 2000), this reduction in ENSO strength during the mid-Holocene (3-5 ka) was a response to orbitally driven seasonality changes. Other authors whose records showed a reduction in ENSO variance during the mid-Holocene suggested that the driver was a change in the equatorial Pacific mean state (Chen et al., 2016; Cobb et al., 2013; White et al., 2018).

Speleothems (secondary cave carbonates) offer a variety of environmentally sensitive proxy records and accurate age control. They have near-global distribution (Fairchild and Baker, 2012) and temporal resolution ranging from intra-seasonal

(Ridley et al., 2015) to orbital (Cheng et al., 2016). Stable oxygen isotope ratios ($\delta^{18}O$) of speleothems can provide
information about regional moisture (e.g., rainfall amount and water vapor source) (Breitenbach et al., 2010; Lachniet,
2009), whereas the interaction of soil $CO_2$ and local infiltration processes inherent to the karst system are reflected by
variations in stable carbon isotopes ($\delta^{13}C$) (Fohlmeister et al., 2020; Genty et al., 2003) and trace-element ratios (e.g., Mg/Ca
and Sr/Ca) (Fairchild and Treble, 2009). Less-used indicators of environmental conditions include the variations in physical
features of speleothems such as density, colour, and lamina thickness, which are a function of hydrology, karst conditions,
drip rate, and drip water chemistry (Fairchild and Baker, 2012). These processes are reflected in the speleothem lamina
growth rate and seasonal depositional changes (dry/wet) and can be observed by using high resolution imaging (Faraji et al.,
2021). Accounting for these features can improve the interpretation of geochemical data.

Here we present a seasonally resolved mid-Holocene multi-proxy stalagmite record from Niue Island in the southwestern
Pacific. This multi-proxy approach integrates an array of stable isotopes, trace elements, and physical properties
measurements. The research objectives of this study include a) characterization of the nature of the speleothem laminae; b)
identification of the environmental controls on the physical and geochemical proxies; c) establishing a chronology for the
proxy records; d) extraction the fundamental periodicities from the observed proxy variability; and e) exploration of
interactions between and possible controls on the main modes of climate variability in the region. We used traditional
statistical analysis such as principal component analysis and wavelet analysis to extract the governing mechanisms of the
climatic variability and advanced statistical tools such as recurrence plots to identify predictable, recurring features within
the time series and detect transitions in climate system dynamics.

We present the first continuous, sub-annually resolved record of past hydrological changes associated with the south/north
movements of the SPCZ between 6.4 ka and 5.4 ka BP and show that ENSO was active during this period. In addition, these
findings provide evidence on the interaction of ENSO and rainfall seasonality in the tropical Pacific and insights into the
effect of El Niño and La Niña events on the predictability of the climate system.

## 2. Geographic and climatic setting

Niue Island is a carbonate edifice located in the southwestern tropical Pacific (19°03' S / 169° 55' W) (Fig. 1). The island
reaches a maximum altitude of ca. 60 m a.s.l. and hosts numerous caves, particularly in the Mutulau reefal limestone and the
coastal cliffs (Aharon et al., 2006). The natural vegetation cover is characterized by dense tropical forest.
Located ca. 490 km north of the Tropic of Capricorn, Niue is characterised by a tropical (Af) climate (Peel et al., 2007), with
a mean annual air temperature of 24°C ($T_{min}$ = 20°C in July, $T_{max}$ = 29°C in January). Niue receives ca. 2000 mm of rainfall
per year, mostly from November to April, while the period from May to October is cooler and relatively dry (Meteorology



and CSIRO, 2011). Rainfall variability is controlled by the seasonal movement of the South Pacific Convergence Zone (SPCZ) (Fig. 1). The southward-positioned SPCZ brings convective rainfall during the warm wet season, while the northward shift of the SPCZ results in cool and drier conditions. Due to its location near the southwestern margin of the SPCZ, Niue is sensitive to sea surface temperature and atmospheric circulation changes linked to interannual El Niño-

Southern Oscillation (ENSO) dynamics (Rasbury and Aharon, 2006). El Niño events are associated with a northeastward displacement of the SPCZ and hence drier conditions in Niue, particularly during the normally wet austral summer season (Fig. 1a). During La Niña events, the SPCZ inclination shifts ca. 1-3° towards the south-east resulting in increased rainfall in Niue (Fig. 1b) (Brown et al., 2020; Lorrey et al., 2012). Between 1969 and 2010, at least one or two TCs have hit Niue each wet season (Meteorology and CSIRO, 2011). During El Niño years, above-average sea surface temperatures (SSTs) in the

central and western equatorial Pacific enforce positive cyclonic vorticity (Vincent et al., 2011), resulting in an increase of frequency of TCs (Meteorology and CSIRO, 2011).

### 3. Material and Methods

#### 3.1 Stalagmite C132

Stalagmite C132 was collected from Mataga Cave, between the villages of Tuapa and Hio on the West coast of Niue Island. The stalagmite was found broken, with the top segment of the stalagmite missing, in a small grotto at the end of the cave. The collected segment is 43.4 cm long (Fig. 2) with visible layers of milky-white/pale porous calcite (PPC) and dark dense calcite (DDC) (Hartland et al., 2014).

#### 3.2 Sampling for geochemical analyses

Powder samples were obtained for geochemical analyses using a Sherline microdrill with a 1 mm diameter tungsten-carbide drill and a digital read-out. Eleven powder samples of ~ 200 mg were collected for U/Th dating. Stable isotope samples were drilled every 3 mm (low-resolution, n=144), and milled every 50 μm (high-resolution, n=5102) following the procedure outlined in Baldini et al., (2021). Fig. 2 shows the sampling tracks.



### 3.3  U-Th dating

The U-Th ages were determined using a Nu Instruments Plasma MC-ICP-MS at the University of Melbourne. Sample powders were dissolved and equilibrated with a $^{229}$Th/$^{233}$U mixed spike solution before U and Th were separated from the matrix using Eichrom TRU-Spec resin. The purified U/Th fraction was introduced to the MC-ICP-MS via a Cetac Aridus membrane desolvator, giving total system efficiencies of ca. 0.3% for both elements. See Hellstrom (2003) for detailed description of the protocol.

### 3.4  Greyscale analyses

Greyscale values were extracted from high resolution (2400 dpi) scans of the stalagmite surface along the growth axis using the image analysis software ImageJ version 1.51k (https://imagej.nih.gov/ij/index.html; Schneider et al., 2012). This analysis provides a record of intensity values between zero (black) and 255 (white), with a spatial resolution of 10.6 μm.

### 3.5  Age-depth modelling

The age-depth model of stalagmite C132 was initially constrained using eight U-Th ages as input to the COPRA age-modelling software (Breitenbach et al., 2012). This U-series-based age-depth model was then further constrained by comparison with the layer counting chronology. The series of counted layers vs. stalagmite depth was anchored to the depth of the topmost U/Th date. This layer chronology fell within the 2σ uncertainties of the older U/Th dates. This procedure leaves an overall error margin of ±20 years. The final age model is the median record based on ensembles of 2000 Monte
Carlo age-depth realisations derived using the COPRA routine (Breitenbach et al., 2012).

### 3.6. Speleothem oxygen and carbon isotope analyses

Every fourth high-resolution sample was analysed, resulting in an actual resolution of 200 μm. First set (n=607) of samples covering the depth ranges 4-16.25 mm and 46.85-92.26 mm was measured at Ruhr University Bochum. Between 90 and 110 μg of sample powder was acidified with orthophosphoric acid at 70°C and reacted for 60 min before analysis. The released
CO$_2$ gas was dried and measured in continuous flow mode on a ThermoFisher MAT253 gas source isotope ratio mass spectrometer coupled to a GasBench II (ThermoScientific, Bremen, Germany). Results are presented in delta notation with δ-values reported as parts per thousand (‰) relative to the international Vienna PeeDee Belemnite (VPDB) standard. Results are corrected using a two-point calibration using the international standards IAEA-603 and NBS18. The long-term 1σ reproducibility of the internal standard is 0.06 ‰ for δ$^{13}$C and 0.09 ‰ for δ$^{18}$O.
A second set (n=740) of isotope measurements, covering the depth ranges of 92.35-205.26 mm and 16.84-47.16 mm, was performed at ETH Zurich using a ThermoFinnigan Delta V Plus isotope ratio mass spectrometer coupled to a ThermoScientific Gasbench II. See technical details in the supplementary material.

### 3.7 Isotope analysis of rain and drip water

Rain and drip water samples from four Niue caves were collected in a fieldwork campaign in February 2020. The oxygen and hydrogen isotope ($\delta^{18}O$ and $\delta D$) composition of samples was measured using cavity ring-down spectroscopy (CRDS; Steig et al., 2014) at the Universidad de Almería, Spain. The CRDS device was interfaced with an A0211 high-precision vaporiser. The internal standards were JRW, BOTTY and SPIT. The results were normalised to the VSMOW (Vienna Standard Mean Ocean Water). Typical long-term instrumental precisions (±1SD) were ±0.06‰ for $\delta^{18}O$ and ±0.6‰ for $\delta D$,

based on the repeated analysis of an internal standard every 6 samples.

### 3.8 Trace element analysis

Concentrations of a suite of 15 elements were measured along the growth axis of C132, following the greyscale track. Measurements were performed at the University of Waikato (New Zealand) by laser ablation-inductively coupled mass spectrometry (LA-ICP-MS) using a RESOlution SE 193 nm excimer laser ablation system equipped with a Laurin Technic

S155 laser ablation cell coupled to an Agilent 8900 QQQ-ICP-MS. See technical specifications at the supplementary material.

## 4. Results

### 4.1 U-Th dating and age modelling

The age model of stalagmite C132 is confined by eight U/Th ages. Two out of the eleven U/Th measurements were

discarded due to large uncertainties, one due to likely hiatus (Table 1). The latter (69.3 ka) in the lowermost part of the stalagmite, was measured on a sample taken below a visible crystal fabric change (Fig. 2). Radiometric dating indicates that the stalagmite grew continuously from 6.428 to 5.411 ka BP (with present referring to 1950 CE). The Holocene portion of the record is the focus here. The Holocene U-series chronology is further refined by layer-counting based on greyscale data. The layer count indicates that the Holocene part of the record spans 1019 years.

**4.2 Greyscale record**

The greyscale intensity values vary between 93.5 and 273.3 (Fig. 3a), with a mean of 179.4. Pale porous calcite (PPC) laminae have higher values than adjacent dark dense calcite (DDC) laminae. The greyscale record has the highest resolution of all C132 proxy records and exhibits variability from seasonal to multi-decadal time scales. The growth rate record reveals a notable change in average growth rate at 6.1 ka BP, with faster growth prior to 6.1 ka BP (0.57 mm/yr), and slower growth

(0.34 mm/yr) between 6.1 ka and 5.4 ka BP (Fig. 3f).





### 4.3 Oxygen and carbon isotopes record

In total, 144 samples were measured at interannual resolution (6.39-6.002 ka BP), and 1347 samples were measured at sub-annual resolution (6.002-5.422 ka BP). Over the analysed interval, $\delta^{18}O$ varied between -7.19 ‰ and -3.47 ‰, with a mean of -5.5 ‰ (Fig. 3d). The $\delta^{13}C$ values ranged from -13.74 ‰ to -6.5 ‰, with a mean of -9.62 ‰ (Fig. 3e). The $\delta^{18}O$ and $\delta^{13}C$

time series record sub-annual to centennial-scale changes.

The $\delta^{18}O$ and $\delta^{13}C$ values are positively correlated (r=0.58, *p*<0.001), with both isotope systems varying synchronously throughout the record (Supplement S1). The stable isotope values fluctuate within ±1.5 σ of the mean. The highest $\delta^{18}O$ values of the record occur between 5.50 and 5.46 ka BP, where a decadal-scale positive excursion of ca. 1.3 ‰ is followed by a rapid decrease towards the mean (Fig. 5e). A similar trend is observed in $\delta^{13}C$, although in this case the values are much

closer to the mean.

### 4.4 Trace elements records

High-resolution LA-ICP-MS analysis showed distinct variations in a suite of 14 elements. The results are reported as metal/Ca ratios and summarised in Table 2. This section focuses on variations of Mg and Zn, which represent contributions from the host rock and the soil, respectively (Fig. 3b and c) (Aharon et al., 2006; Murgulet, 2010).

The resolution of the LA-ICP-MS analyses allows the detection of sub-annual variability in the trace element record (Tables 3 and 4 in sec. 5.1). Mg/Ca concentrations vary between 2.95 and 36.07 mmol/mol with a mean value of 11.92 mmol/mol (Fig. 3b). The Zn/Ca ratios ranges from 0.1 to 420 μmol/mol with a mean value of 17.95 μmol/mol (Fig. 3c). Higher Zn/Ca values generally coincide with PPC laminae (Fig. 4e). Technical difficulties resulted in a gap in the trace element data at 276.225-281.662 mm (6.219-6.213 ka BP).


### 5.    Statistical analysis

### 5.1 Principal component analyses (PCA)

Principal component analysis (PCA) allows the identification of associations which might be interpreted as common forcing in the proxy time series. We carried out several PCAs on different groupings of datasets derived from stalagmite C132. All

records included in each PCA were standardised to the lowest resolution in the group by averaging the data corresponding to each lowest-resolution time interval to accommodate the differences in temporal resolution (Table 3). Table 4 summarises the different PCA groupings and temporal resolution. Data preparation prior to principal component analyses included imputing missing values using the iterative PCA algorithm from the missMDA library in the software R and log transformation of the data series. PCAs were performed in R software using the PCAshiny algorithm from the Factorshiny

(v.2.2) package.



Three PCAs were performed: PCA-1, PCA-2, and PCA-3 (Table 4). Technical difficulties resulted in a gap in the trace element data at 276.225-281.662 mm (6.219-6.213 ka BP). Below this gap, several of the trace element series have a substantially higher variance compared to those above the gap. To avoid potential instrumental biasing, the trace element time series data was split into two sets and PCAs were performed separately on the uppermost and lowermost sections

(PCA-1a and PCA-1b). PCA-3, including only trace elements, is presented in the Supplement (S2).

Fig. 5 displays the PCA results, where each dot signifies the loading of the respective record onto principal components (PCs) 1 and 2. For PCA-1a, the first two principal components (PCs) explain 56.48% of the variance in the original data (Fig. 5a). Zn, Mn, Fe, Pb, and Al show a strong positive correlation with PC1 (loadings > 0.7) and a moderately positive correlation with PC2 (loadings < 0.6). Mg, U, Sr, and P define a group which is moderately negatively correlated with PC2

(loadings < 0.6) and weakly to moderately positively correlated with PC1 (loadings < 0.5). The first two components of PCA-1b explain 61.64% of the total variance and show similar associations of elements (with Br added to the second group) and similar correlations to the two PCs as in PCA-1a.

PCA-2 produces similar groupings to PCAs 1a and 1b but also indicates a third group that includes Ba, Na, and Br (green circle, Fig. 5c) and is moderately positively correlated to both PC1 and PC2 (loadings < 0.6). The stable isotopes do not

contribute to PC1 and PC2; instead, they present high loadings for PC4. In both PCA-1 and PCA-2 the greyscale record is projected onto the lower left quadrant of the coordinate system spanned by the PCs, indicating an inverse relationship to the group of elements Sr, Mg, U, and P. Full-resolution PCAs of trace element data (PCA-3a and b) were also performed with similar results (Supplement S2).

### 5.2 Seasonality determination

For the study of mid-Holocene rainfall seasonality, we performed a minor recalibration on the monthly-scale dating of the greyscale record. Based on the fact that mean rainfall is higher during the wet season (November to April) than in the dry season (May to October), the greyscale record was anchored on its time axis such that wet season averages are higher than dry season averages in the maximum possible number of years. This yielded a (constant) shift of +5 months. Rainfall seasonality was afterwards calculated for each year as the difference between average greyscale values in the wet and the dry

season (Fig. 7a/b and Supplement S3).

### 5.3 Spectral analysis

To investigate which processes drive the variability recorded in the C132 dataset, we applied the biwavelet package in R to perform wavelet power spectrum analyses using Morlet wavelets (Torrence and Compo, 1998) on the suite of annually resolved proxy data, the seasonality record and the two first principal components (PCs) derived from PCA-2.

The wavelet spectrum of the greyscale record displays significant power (>95% confidence level) in the ENSO band (2 to 8 years) continuously through the recorded period (Fig. 6a). The seasonality record exhibits episodic ENSO-scale variability with two periods of muted ENSO activity, the first from 6030 to 5900 y BP and the second from 5600 to 5500 (Fig. 6b).



The wavelet spectra of PC1 and PC2 of PCA-2 show irregular patches of significant periodicities associated with ENSO-band variability at 6000-5950, 5700-5650, and 5500-5400 years BP (Supplement S4).

## 5.4 Recurrence analysis

Recurrence analysis is used to test if a time series revisits formerly visited states in a regular or erratic fashion. The predictability of rainfall seasonality was computed based on a recurrence plot (RP) analysis (Marwan et al., 2007; Westerhold et al., 2020). RPs with a fixed recurrence rate of 15% were computed on 200-year sliding windows after embedding each time series segment with an embedding dimension of 3 and an embedding delay of 2; the selection of these parameters is explained in the Supplement S5.

## 6. Discussion

### 6.1 Interpretation of environmental proxies

### 6.1.1 Greyscale values

The greyscale variability of stalagmite C132 is a measure of the alternation between PPC and DDC laminae, which are related to crystal growth and matrix-density variations. Higher greyscale values represent porous crystal growth arrangements, whereas lower greyscale values reflect denser calcite crystal patterns (Fig. 4a). Factors such as drip water saturation, drip water pH, drip rate, and $CO_2$ degassing promoted by cave ventilation govern the formation of distinctive crystal fabrics depending on the seasonal environment (Baker et al., 2008; Frisia, 2015).

We interpret the variation in crystal growth style (and thus greyscale values) as a function of the dissolved carbonate supply. During the wet season, the supply of drip water (and with it dissolved inorganic carbon, DIC) is high, allowing for rapid $CaCO_3$ deposition. In the dry season, water supply is more restricted, drip rates are lower, and thinner, denser, and darker carbonate laminae form. Consequently, as previously observed in other coastal Niuean caves (Aharon et al., 2006; Rasbury and Aharon, 2006), long crystals form in the wet season and micritic carbonate (i.e., micro-crystalline, nearly glassy) layers form during the dry season.

We suggest that the mechanism that promotes the preservation of seasonal signals in the C132 proxy records is the seasonal cycle in drip water supersaturation, leading to sub-annual changes in stalagmite growth rate. These factors produce seasonal variation in geochemical proxy records in fast-growing speleothems (Carlson et al., 2018; Hartland et al., 2014).

### 6.1.2 Oxygen isotope values of rain, drip water and speleothem calcite

We attribute the sub-annual $\delta^{18}O$ variability observed in the C132 record to changes in amount of rainfall delivered to Niue Island over the depositional period. Due to the small size of the island and its geographical location, the source of Niue's rainfall is entirely oceanic. The summerly southward movement of the SPCZ and tropical cyclones bring strong vertical





convective rainfall with a depleted $\delta^{18}O$ signature (see for example Supplement S6). Rain and drip water from Niue Island fall on the South Pacific Meteoric Water Line (SPMWL: $\delta D = 7.7 * \delta^{18}O + 9.3$, $r^2=0.96$) derived from Samoa and Rarotonga rainfall databases (IAEA/WMO, 2001), as well as on the GMWL (Supplement S5). These results are consistent with
monitoring studies of two other Niuean caves that suggest seasonal variability in the isotopic composition of drip waters, with higher $\delta^{18}O$ values in the dry season and lower $\delta^{18}O$ values during the wet season (Murgulet, 2010; Rasbury and Aharon, 2006; Tremaine et al., 2016). Changes in stalagmite $\delta^{18}O$ thus provide information about the location of the SPCZ and/or the prevalence of tropical cyclones in the Central Pacific during the mid-Holocene. Lower stalagmite $\delta^{18}O$ values are found in the PPC laminae indicating the intensity of the wet season and higher $\delta^{18}O$ values found in the DDC that represent
the dry season.

### 6.1.3 Carbon isotopes of speleothem calcite

The $\delta^{13}C$ values of stalagmite C132 have a similar range to those of a stalagmite record from the nearby Avaiki Cave (Murgulet, 2010). In both records, the $\delta^{13}C$ values are lower in PPC laminae deposited during the wet season and higher in
the DDC laminae formed during the dry season (Fig. 4f). As with the greyscale changes discussed above, the variations in the isotopic composition of the alternating laminae can be explained by seasonal changes in water and dissolved inorganic carbon (DIC) supply. It has been shown that $\delta^{13}C$ values in drip waters in Niue caves represent a mixed signal from both soil $CO_2$ (-29.4 ± 0.09 ‰; 91%) and carbonate bedrock (-0.4 ± 0.09 ‰; 9 %) (Aharon et al., 2006). The $\delta^{13}C$ variations in C132 are likely caused by changes in the proportions of these two carbon sources. Lower drip rates during the dry season allow for
prolonged $CO_2$ degassing from the dripping water, enhancing kinetic fractionation and resulting in higher speleothem $\delta^{13}C$ values. During the wet season, increased moisture supply promotes microbial and vegetation activity in the soil (Cruz et al., 2005; Genty et al., 2003), decreasing the pH of the infiltrating water and intensifies water-rock interaction (McDermott, 2004) while reducing prior calcite precipitation (PCP), resulting in lower speleothem $\delta^{13}C$ values.

### 6.1.4 Trace elements

A series of PCAs were used to investigate the processes controlling trace element variations in stalagmite C132 at annual and sub-annual scale. All PCAs consistently reveal two main groups based on their loadings on PC1 and PC2: Zn, Fe, Cu, Pb, Al, and Mn (Group 1) and U, Sr, Mg, and P (Group 2) (Fig. 5). Group 1 includes the soil-derived elements which are transported into the epikarst via high infiltration events (Borsato et al., 2007; Hartland et al., 2012; Oster et al., 2017). Group
2 comprises elements that are derived from the host-rock as well as variable inputs of marine aerosol. This includes Mg and Sr, which are incorporated into the speleothem calcite by water-rock interaction (WRI) and/or modified by prior calcite precipitation (PCP) and are often interpreted as a proxy for local hydrology, with high Mg and Sr indicating drier conditions





(Fairchild and Baker, 2012). It is likely that in this cave system, the incorporation of U and P into the calcite follows a similar mechanism as observed for Mg, i.e., substitution for $CO_3^{2-}$ in the crystal lattice. Dry periods induce less host-rock dissolution, leading to lower $CO_3^{2-}$ activity and increased U and P partitioning (Wynn et al., 2018), allowing higher incorporation of U and P in the slower-growing DDC laminae. In contrast, supersaturation in the wet season promotes a higher growth rate through enhanced degassing and increased competition for U and P ions to replace carbonate in the lattice of the PPC laminae (Jamieson et al., 2016).

Group 2 (host-rock) elements are strongly positively correlated with PC2, which explains ca. 26% of the total variance. Importantly, the greyscale record is moderately to strongly negatively correlated with PC2, while Group 1 (soil-derived) elements are weakly to moderately negatively correlated with PC2. Since greyscale values are primarily controlled by drip water (and thus rainfall) amount, PC2 likely represents the variation between wetter and drier years, with the positive direction indicating dry and the negative direction indicating wet (see note to Fig. 5). Wetter conditions would result in more soil-derived elements being transferred to the cave, leading to the negative correlation with PC2, while host-rock elements are concentrated in the calcite during drier conditions due to lower drip rates.

The same control (i.e., drier conditions) would also serve to concentrate Mg derived from marine aerosols within the epikarst water store, leading to higher Mg/Ca and other element/Ca ratios that are highly concentrated in seawater (e.g., Na, Sr, Ba). It is notable that these elements are separated from the soil-derived elements by PC2 and distributed toward the host-rock-derived elements. Thus, while increases in host-rock-derived elements (e.g., U, Mg) support an interpretation of drier climate states, the similar result would be found if a small contribution of marine aerosol were present. Although this study does not draw on drip water monitoring, it is highly probable that marine aerosol also contributed to the elemental composition of C132, similar to earlier findings (Tremaine et al., 2016). This reasoning does not affect the general interpretation of the elemental data.

PC1 explains 28-35% of the variance in the datasets. Soil-derived elements are strongly positively correlated with this axis, while host-rock/marine elements show a weakly positive to no correlation, and greyscale shows weakly negative to no correlation. Soil-derived elements are transferred to the cave by the flow of water through the epikarst. However, overall wet/dry conditions would affect greyscale and host-rock derived elements to a greater and more predictable extent, as discussed above. A major transfer of soil-derived elements may occur through flushing of the epikarst during extreme rainfall events (Hartland et al., 2012), such as tropical cyclones. These events are short-lived and would be unlikely to have a major effect on the style or rate of (longer-term) calcite crystallisation, thus having a limited effect on greyscale and host-rock elements. We thus interpret PC1 as controlled by the prevalence of extreme rainfall events in the tropical Pacific.

With loadings of comparable magnitude on both PC1 and PC2, Na and Br are located between the host-rock and the soil-derived trace metal groups. As previously noted, Na and Br are concentrated in seawater, and previous studies estimated a seawater contribution of 89% for Na and Br in the drip water of Niuean coastal caves (Tremaine et al., 2016). Following this finding, we interpret these elements as being mostly derived from marine aerosols.



## 6.2 Climatic interpretation of the proxy time series

The term 'rainfall seasonality' is used here to refer only to the difference in the amount of rain between the wet and dry seasons within one annual cycle, as measured by the difference in the greyscale values between a PPC lamina and the
adjacent DDC lamina. We developed a seasonality time series based on the difference between the PPC lamina peaks (wet season maximum) and the DDC lamina troughs (dry season maximum) from the greyscale record (Supplement S3). Low rainfall seasonality values refer to low contrast between the wet and dry seasons, while high rainfall seasonality values represent higher contrast between wet and dry seasons. Changes in the consistency of seasonal oscillations are defined by the DET parameter, which allows us to quantify seasonal predictability in the system and to identify transition points from
predictable to random conditions and *vice versa* (Fig. 7g). Periods of more predictable seasonality are distinguishable from those with irregular seasonal variations based on the 95%-confidence level obtained from a bootstrapping procedure (dotted grey line, Fig. 7g). Whenever this confidence level is exceeded, seasonal rainfall variations have significantly shifted towards a regime of greater or lower predictability.

## 6.3 The Mid-Holocene hydrological variability of Niue Island

Our 1019-year hydrological reconstruction from Niue Island reveals five main stages delineated by multi-centennial (200-300 yrs) oscillations between generally increased and decreased predictability of rainfall seasonality during the mid-Holocene. Phase 1, from 6.4 ka to 6.2 ka BP, is characterized by wetter conditions as indicated by a high mean growth rate and greyscale (Fig. 3a, f). PC2 (background rainfall) is relatively high during this period, while PC1 (indicating extreme events, possibly driven by strong ENSO) is moderate to low, though with some high peaks. During this period the wavelet
spectrum of the greyscale record (overall rainfall) shows significant continued ENSO band variability, and the wavelet spectrum of the seasonality record present also periods of significant ENSO band variability but sparse on time (Figs. 6 and 7f).

In Phase 2 (6.2-5.9 ka BP), the variability in background rainfall is reduced below the mean, and heavy rainfall/high infiltration events increased in frequency (Fig. 7a, b). During most of this period seasonality values are moderate to low,
with increased predictability of rainfall seasonality (Fig. 7e, g). This suggests that during Phase 2 rainfall more evenly spread out over the year, presumably due to high infiltration events in the dry season linked to TCs.

Phase 3 (5.9 to 5.72 ka BP) starts with the lowest PC2 values (background rainfall), occurring after a notable decrease at 5.85 ka BP. After ca. 5.8 ka BP the PC2 returns to higher values again, suggestive of "normal" background rainfall. Concurrently, the low PC1 signal suggests few high infiltration events. Two peaks in $\delta^{18}O$ and $\delta^{13}C$ records coincide to the
"normal" background rainfall conditions reveal two decadal-scale maxima. It seems that Phase 3 was generally wet, with superimposed decadal-scale dry episodes. We interpret these signals as indication for pronounced seasonality and low rainfall predictability due to significant influence of ENSO on atmospheric conditions over Niue.



Phase 4 (5.72 to 5.5 ka BP) is characterised by multi-decadal oscillations between wet/dry periods, represented by background rainfall (PC2), the high infiltration events record (PC1), as well as the $\delta^{18}O$ and $\delta^{13}C$ records. The most

pronounced drying signal in the $\delta^{18}O$ record occurs around 5.5 ka BP at the same time there is a low in background rainfall, this could be indicating a northward displacement of the SPCZ. Seasonality is quite stable except for a decades-long maximum at ca. 5.65 ka that coincides with a decrease in the background rainfall and high infiltration events. This peak also corresponds to a shift from consistently significant ENSO power in the greyscale record to ENSO-band peaks in the seasonality record. Phase 4 thus was characterized by predictable seasonal rainfall, except for the short interlude at 5.65 ka

BP.

In the final Phase 5 of our record (5.5 to 5.4 ka), the background rainfall rapidly increases while the high infiltration events record shows a more subdued and gradual rise. This increase toward wetter conditions is supported by low $\delta^{18}O$ values. During this period, ENSO variance is only marginally significant in the greyscale record, while it is muted in the seasonality record. Predictability of rainfall seasonality is low and decreasing.

Significant ENSO-band variability in the greyscale record occurs at different times than in the seasonality record (green and yellow peaks, respectively, in Fig. 7f), which suggests different controls. Since the seasonality record reflects the wet/dry season contrast, whereas the greyscale record reflects mostly the wet season, we suggest that the ENSO-band variability in seasonality is controlled by changes in the amount of rain during the dry season, which would increase or reduce the contrast between seasons. El Niño conditions are characterised by drier conditions than normal; however, TCs are more frequent in

El Niño years (de Scally, 2008). Given that dry season rainfall is generally relatively low, overall background reduction in dry season rainfall during El Niño years may be counterbalanced by an increase in TC occurrence. This would lead to a reduced seasonality (drier wet season and wetter dry season) during El Niño years and decrease predictability.

In Niue Island the most important controls on rainfall seasonal variability are the location of the SPCZ, followed by ENSO. During El Niño events, the wet season is drier, thereby reducing the contrast between the wet and dry seasons (i.e., reduced

rainfall seasonality). La Niña events bring wetter rainy seasons, thus increasing the seasonal contrast (i.e., amplified rainfall seasonality). On a visual basis, the C132 record shows a consistent positive correlation between greyscale/seasonality ENSO band variability and rainfall seasonality predictability. In general, when ENSO band variance is significant, there is a decrease in the predictability of seasonality, i.e., the system turns into a more stochastic conditions, whereas when ENSO band variance is reduced/muted the seasonality predictability is increased, and individual dips in predictability often

correlate with significant ENSO band power in seasonality (e.g., at 6.3, 6.03, 5.88, 5.75, and 5.6 ka BP). Assuming that La Niña events bring wetter conditions only during the wet season, because December typically the peak of La Niña events (Timmermann et al., 2018), this stretch the seasonal cycle above normal. It would be very unlikely that La Niña events result in wetter dry season because usually dry season months (June-August) correspond to either El Niño to La Niña transition or early growth El Niño stages; hence the relationship between ENSO activity and rainfall seasonality in Niue can be

confirmed. A similar positive relationship between the seasonal cycle amplitude of near-equatorial sea surface temperatures (SSTs) and ENSO band variance was observed in Holocene coral records from the central Pacific (Emile-Geay et al., 2016).



The initial oceanic conditions for the development of a La Niña events depend of multiple factors such as spatial-temporal heat dynamics in the equator, the precedent El Niño flavour (East Pacific or Central Pacific El Niño events) (Kessler, 2002), these changing conditions reduce the predictability of La Niña events, although this conclusion is based in an analysis that
assess the forecasting skills of La Niña events and not the seasonal cycle (Timmermann et al., 2018), is similar to our findings in the way that La Niña events result from destabilization of the atmosphere-climate system and therefore less predictable conditions.

Our findings suggest that ENSO variability modulates the amplitude of the seasonal rainfall cycle not only at an inter-annual scale, but also multi-decadal and centennial scales. These lower-frequency oscillations could in turn be modulated by long-
term stationary states of the location of the SPCZ as well as its spatial configuration, which is controlled by the feedback of internal variability factors (Brown et al., 2020). This ENSO and tropical Pacific mean state connection at decadal and centennial scales has been detected in other highly resolved mid-Holocene ENSO records from the central Pacific (Cobb et al., 2013) and Borneo (Chen et al., 2016). Niue record contribute with new evidence of the interaction between SPCZ, ENSO and the mean state of the tropical Pacific throughout the identification of predictable and non-predictable stages in the
climate system.

## 7. Conclusions

Investigated stalagmite from Niue Island in the southwestern Pacific offers a seasonally resolved multi-proxy reconstruction of mid-Holocene (6.4 to 5.4 ka BP) rainfall changes associated with dynamics of the South Pacific Convergence Zone (SPCZ).

The combination of U-Th dating and layer counting allows for constructing an accurate chronology for the multi-proxy record, while the use of the non-destructive greyscale analysis supports and strengthens the interpretation of the geochemical proxies.

Wet/dry conditions controlled by seasonal shifts of the SPCZ are recorded in the petrography, trace element distribution, and isotopic composition ($\delta^{13}C$ and $\delta^{18}O$) of the calcite laminae couplets of stalagmite C132. The wet season is reflected in pale
porous calcite (PPC) laminae which are characterised by lower Mg/Ca, Sr/Ca, and U/Ca ratios as well as lower $\delta^{13}C$ and $\delta^{18}O$ values. In contrast, the dry season is reflected by higher Mg/Ca, Sr/Ca, and U/Ca ratios and higher $\delta^{13}C$ and $\delta^{18}O$ values in dark dense calcite (DDC) laminae. We suggest that the physicochemical variations in the stalagmite laminae are modulated by kinetic fractionation forced by differences in drip rate and thus stalagmite growth rate between the wet and dry seasons, which in turn depend on local climatic dynamics.

Within the range of elements studied in sample C132, we have identified two groups due to their source and mechanism of incorporation into calcite show high climatic sensitivity. Group 1 comprises soil-derived elements (Zn, Mn, Fe, Al, and Pb) indicative of high infiltration events resulting from extreme but short-lived rainfall events (e.g., from tropical cyclones) that

lead to significant soil flushing. The second group includes host-rock derived elements (Mg, Sr, U, and P) that are incorporated into the speleothem via water-rock interaction, with further contributions by marine aerosols. These elements

are also sensitive to prior calcite precipitation during periods of reduced infiltration and record hydrological changes at a seasonal scale. This group also records lower baseline rainfall during El Niño events and higher baseline rainfall during La Niña events.

In-depth time series and wavelet analyses of our 1000-year record not only reveals SPCZ changes but provides important insights into ENSO activity during the mid-Holocene. The wavelet analysis suggests that ENSO was continuously active

over the covered period from 6.4 to 5.4 ka BP. However, ENSO affects dry season background rainfall and seasonal rainfall contrast in different ways. We suggest the main effect to be changes in the overall background rainfall, with La Niña years leading to wetter conditions and El Niño years to drier. A secondary effect, related to increased tropical cyclone activity in the dry season during El Niño years, is superimposed on these general dynamics and results in reduced seasonality (wetter dry season and drier wet season). Importantly, tropical cyclone activity linked to El Niño links seasonal predictability with

ENSO-band variability in overall background rainfall, with increased ENSO-band variability corresponding to lower seasonal predictability.

**Author contributions.** CN conducted fieldwork, analysed the data, performed statistical analysis, generated figures, and prepared the original draft of the paper. TB helped to perform statistical analysis and generate figures, BF helped with the

statistical analysis and discussion, AH provided stalagmite sample, conducted fieldwork, and contributed to the discussion. OK conducted fieldwork, contributed to the discussion and editing process. CP contributed to the editing process. SH conducted fieldwork and editing processes. JH performed the U-Th dating. FG performed geochemical analysis on drip water samples. SB, MJ, and AF performed geochemical analysis on speleothem samples. NM contributed to the discussion. AI contributed to the editing process, SFMB supervised the study, conceptualization, conducted fieldwork, contributed to the

results interpretation, and preparation of the paper.

**Competing interests.** The authors declare that they have no conflict of interest.

**Acknowledgments.** We sincerely thank the Niuean Government and landowners for their generous permission and support

of our fieldwork.

**Financial support.** This research has been supported by the European Union's Horizon 2020 Research and Innovation programme through a Marie Skłodowska-Curie grant (no. 691037) and aligned funding from Te Apārangi Royal Society of New Zealand (grant no. RIS-UOW1501), and the Rutherford Discovery Fellowship programme (grant no. RDF-UOW1601).

Cinthya Nava-Fernandez acknowledges financial support from the German Academic Exchange Service (DAAD). Dr. Fernando Gázquez was financially supported by the "HIPATIA" research program of the University of Almería.



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





**Table 1.** Results of the [230]Th/U dating.

| Sample ID | Depth[mm] | U [ngg⁻¹] | [²³⁰Th/²³⁸U][a] | [²³⁴U/²³⁸U][a] | [²³²Th/²³⁸U] | [²³⁰Th/²³²Th] | Age (ka BP)[b] | [²³⁴U/²³⁸U]ᵢ[c] |
|---|---|---|---|---|---|---|---|---|
| C132-1 | 3.5 ± 0.5 | 269 | 0.0533 ± 0.0004 | 1.0767 ± 0.006 | 0.000174 ± 0.000002 | 306 | 5.439 ± 0.059 | 1.0779 ± 0.0061 |
| C132-2 | 29.7 ± 0.5 | 370 | 0.0537 ± 0.0007 | 1.075 ± 0.0027 | 0.000044 ± 0.000001 | 1225.7 | 5.514 ± 0.077 | 1.0762 ± 0.0028 |
| C132-3 | 78.7 ± 0.5 | 311 | 0.0549 ± 0.0005 | 1.0733 ± 0.002 | 0.000026 ± 0.000001 | 2115.2 | 5.653 ± 0.055 | 1.0745 ± 0.0020 |
| C132-4 | 136.3 ± 0.5 | 592 | 0.0568 ± 0.0005 | 1.0793 ± 0.002 | 0.000010 ± 0.000001 | 5768.5 | 5.825 ± 0.059 | 1.0806 ± 0.0020 |
| C132-5 | 161.2 ± 0.5 | 396 | 0.0571 ± 0.0003 | 1.0738 ± 0.0059 | 0.000057 ± 0.000001 | 1002.2 | 5.879 ± 0.047 | 1.0751 ± 0.0060 |
| C132-6 | 185.4 ± 0.5 | 507 | 0.0572 ± 0.0004 | 1.075 ± 0.0024 | 0.000011 ± 0.000001 | 5165.7 | 5.892 ± 0.045 | 1.0763 ± 0.0024 |
| *C132-7 | 213.4 ± 0.5 | 492 | 0.0621 ± 0.0006 | 1.0769 ± 0.0024 | 0.000013 ± 0.000001 | 4948.9 | 6.406 ± 0.066 | 1.0814 ± 0.0018 |
| C132-8 | 290.7 ± 0.5 | n. a. | 0.0604 ± 0.0004 | 1.0799 ± 0.0017 | 0.000105 ± 0.000002 | 575.4 | 6.195 ± 0.044 | 1.0814 ± 0.0018 |
| C132-9 | 365.8 ± 0.5 | 693 | 0.0619 ± 0.0007 | 1.0747 ± 0.0027 | 0.000068 ± 0.000001 | n. a. | 6.39 ± 0.079 | 1.0761 ± 0.0027 |
| *C132-10 | 394.9 ± 0.5 | n. a. | 0.0607 ± 0.0004 | 1.0761 ± 0.0011 | 0.000068 ± 0.000001 | n. a. | 6.256 ± 0.041 | 1.0774 ± 0.0011 |
| C132-11 | 401.4 ± 0.5 | 200 | 0.5210 ± 0.0025 | 1.0981 ± 0.0061 | 0.000091 ± 0.000001 | 5754.3 | 69.264 ± 0.73 | 1.1193 ± 0.0072 |

[a] Activity ratios determined at the University of Melbourne after Hellstrom (2003).

[b] Age in kyr before 1950 AD corrected for initial [230]Th using eqn. 1 of Hellstrom (2006), the decay constants of Cheng et al. (2013) and [²³⁰Th/²³²Th]ᵢ of 0.43 ± 0.043

[c] Initial [²³⁴U/²³⁸U] calculated using corrected age

* Outlier not used in age-depth model

2σ uncertainties in brackets are of the last two significant figures presented.

**Table 2.** Descriptive statistics for the trace element ratios determined along the growth axis of stalagmite C132.

| Metal/Ca ratio | Mean value | SD | Maximum value | Minimum value | Units |
|---|---|---|---|---|---|
| Mg/Ca | 11.92 | 4.95 | 36.07 | 2.95 | mmol/mol |
| Sr/Ca | 0.81 | 0.11 | 1.32 | 0.46 | mmol/mol |
| P/Ca | 1.32 | 0.39 | 3.18 | 0.16 | mmol/mol |
| Al/Ca | 0.04 | 0.11 | 2.499 | 0.00 | mmol/mol |
| Na/Ca | 1.13 | 1.19 | 16.85 | 0.13 | mmol/mol |
| Br/Ca | 1.33 | 0.48 | 3.27 | 0.26 | mmol/mol |
| U/Ca | 0.20 | 0.06 | 0.42 | 0.00 | µmol/mol |
| Pb/Ca | 0.23 | 0.24 | 2.76 | 0.00 | µmol/mol |
| Fe/Ca | 22.6 | 51.5 | 1151.18 | 0.00 | µmol/mol |
| Mn/Ca | 0.96 | 1.9 | 65.70 | 0.00 | µmol/mol |
| Ba/Ca | 1.17 | 0.41 | 4.12 | 0.27 | µmol/mol |
| Cu/Ca | 9.7 | 21 | 393.67 | 0.36 | µmol/mol |
| Ni/Ca | 3.3 | 3.5 | 31.33 | 0.07 | µmol/mol |
| Zn/Ca | 17.95 | 25 | 420.45 | 0.10 | µmol/mol |




**Table 3**. Resolution and time span of the C132 proxy records.

| Proxy | Resolution (points/year) | Time interval (years BP) |
|---|---|---|
| Greyscale (GS) | 30-60 | 5411 – 6428 |
| Trace elements | 8-36 | 5411 – 6428 |
| Stable isotopes high resolution | 2-4 | 5422 – 6002 |
| Growth rate (GR) | 1 | 5412 – 6428 |

**Table 4.** PCA groups and time span. GS = greyscale, GR = growth rate.

| PCA | Groups of proxy records | Resolution of PCA | Time interval (years BP) |
|---|---|---|---|
| PCA- 1a | GS, GR, Trace elements | Annual | 5411 – 6213 |
| PCA- 1b | GS, GR, Trace elements | Annual | 6219 – 6428 |
| PCA- 2 | GS, GR, Trace elements, Isotopes | Annual | 5422 – 6002 |
| PCA- 3a | Trace elements (see Supplement) | Sub-annual | 5411 – 6213 |
| PCA- 3b | Trace elements (see Supplement) | Sub-annual | 6219 – 6428 |




**Figure 1: Austral summer (December-February) daily precipitation across the Pacific. Position of the South Pacific Convergence**
**Zone (yellow lines) during two strong ENSO events: a) El Niño 1998; and b) La Niña 2011. The yellow diamond indicates the**
**location of Niue Island. NOAA Climate Data Record (CDR) of GPCP Satellite-Gauge Combined Precipitation.**







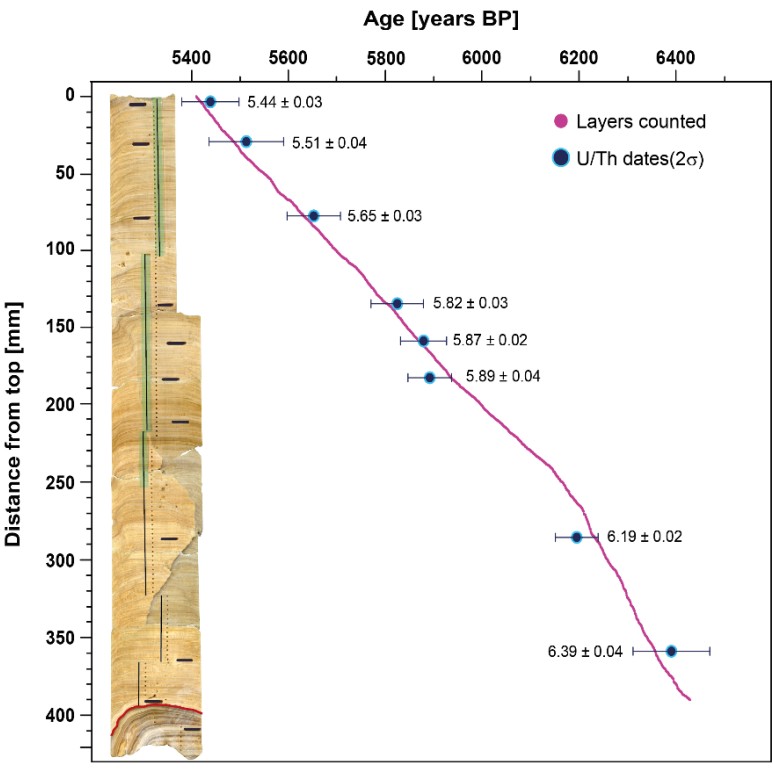

**Figure 2: Age-depth model of the Holocene section of stalagmite C132. Blue circles indicate $^{230}$Th/U-ages with their ±2σ errors. The purple line indicates the layer counting profile. The image of sample C132 shows the U-Th sampling locations (black bars), LA-ICP-MS tracks and greyscale (black lines), the sampling trench for high-resolution stable isotopes (green shading), and low-resolution stable isotope sample locations (dots).**





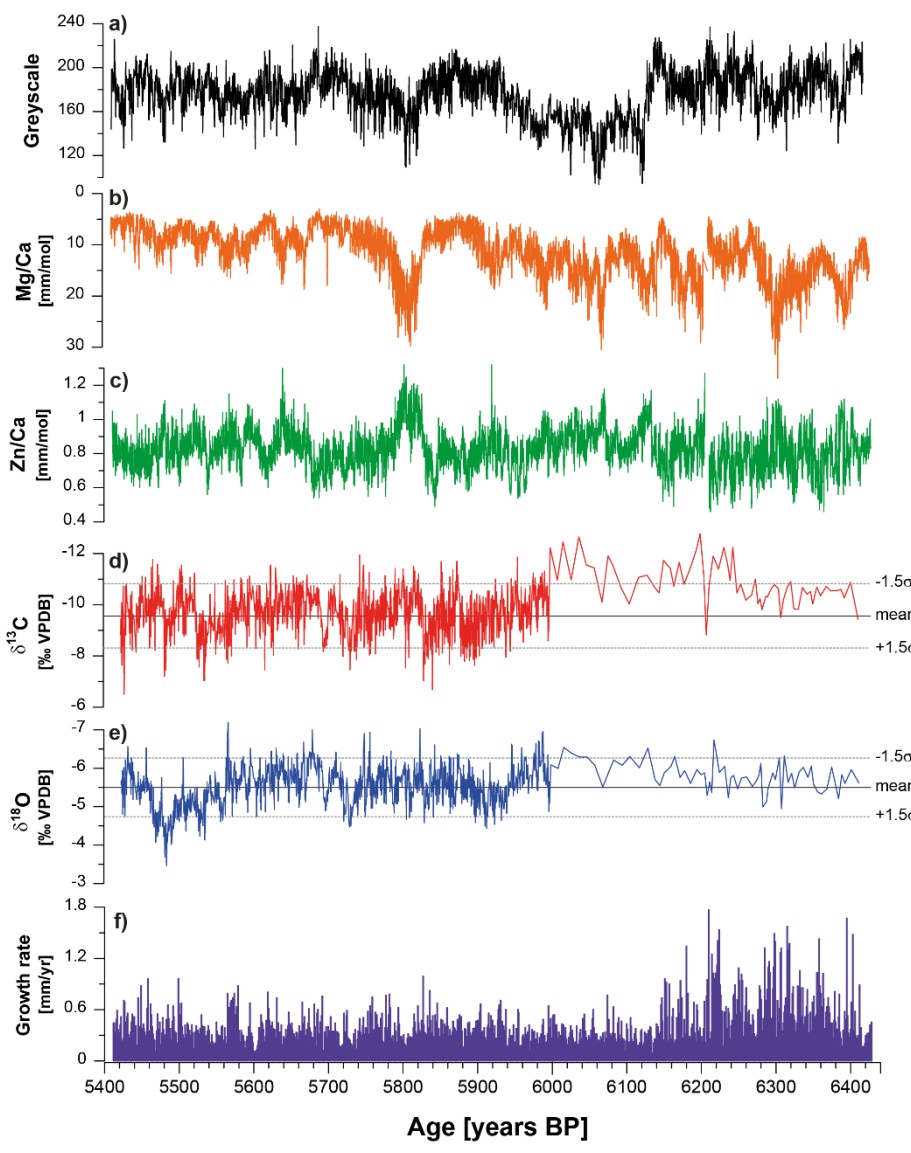

**Figure 3: Proxy time series obtained from stalagmite C132. Note the different temporal resolutions achieved for carbon (d) and oxygen (e) isotope analyses for the periods 6.002-5.422 ka BP (high-resolution) and 6.39-6.002 ka BP (low-resolution).**





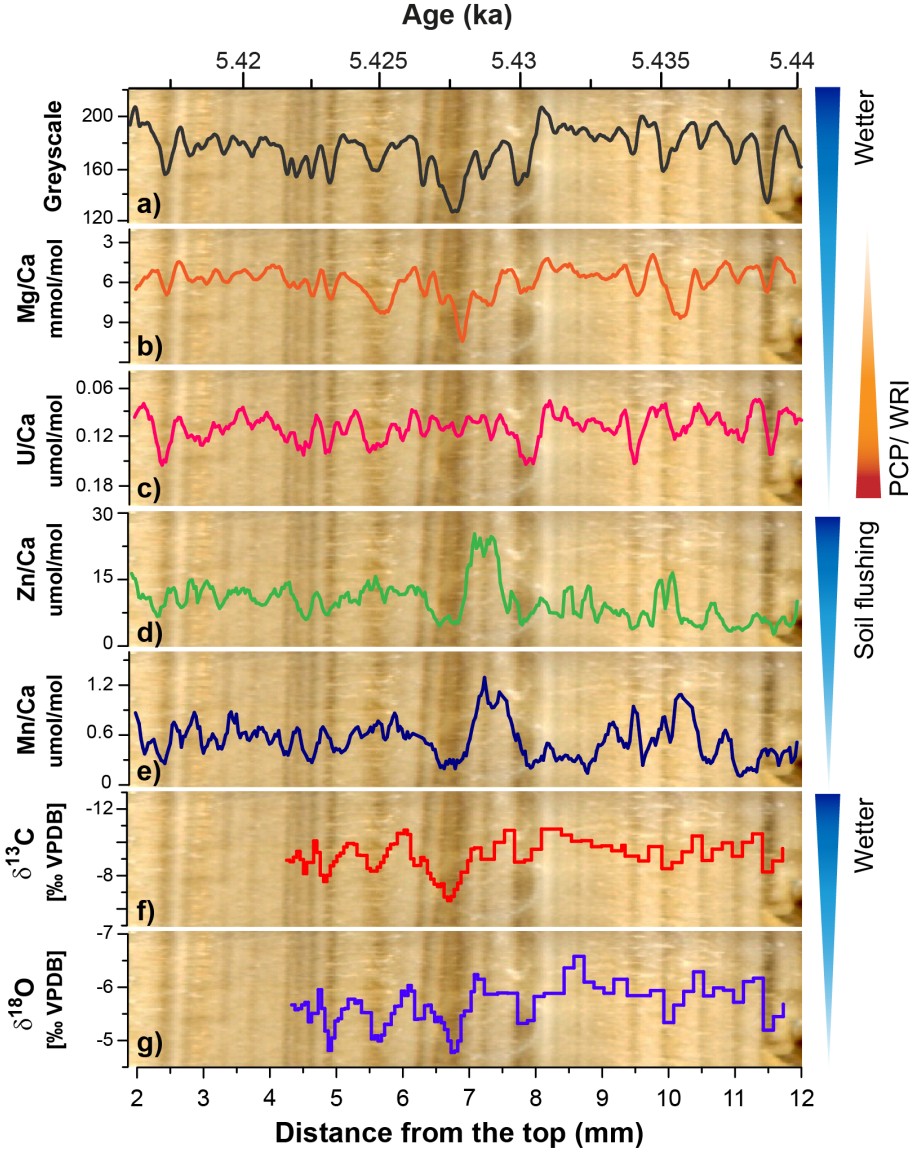

**Figure 4: Detailed view of a 10 mm section of stalagmite C132 with proxy records superimposed on the stalagmite image, showing the relationship of the proxies with the alternation between pale porous calcite (PPC) and dark dense calcite (DDC) laminae. For the explanation of the arrows on the right-hand side, see the discussion section.**






**Figure 5: Results of the principal component analyses. a) PCA-1a, b) PCA-1b, c) PCA-2. All PCAs reveal two prominent groups: group 1 (blue shaded area) is formed by Zn, Mn, Fe, Pb, and Al, and group 2 (orange shaded area) that includes Sr, Mg, U, and P (except for PCA-2 where P is not included in this group). For all PCAs the data were pre-treated assuming normal distributions after a log-transformation (see Section 3.5). For details of the interval and resolution of the datasets for each PCA, see Table 4. Note that in PCA-1a (a), the algorithm assigned the opposite sign to the PC2 axis compared to PCA-1b and PCA-2. This axis is thus reversed for easier comparison.**





**Figure 6: Wavelet spectral analysis of C132. a) Annually resolved greyscale record and b) Constructed seasonality record. Significant (>95%) power is delineated by black contours. The ENSO band (2-8 years) is outlined in red.**






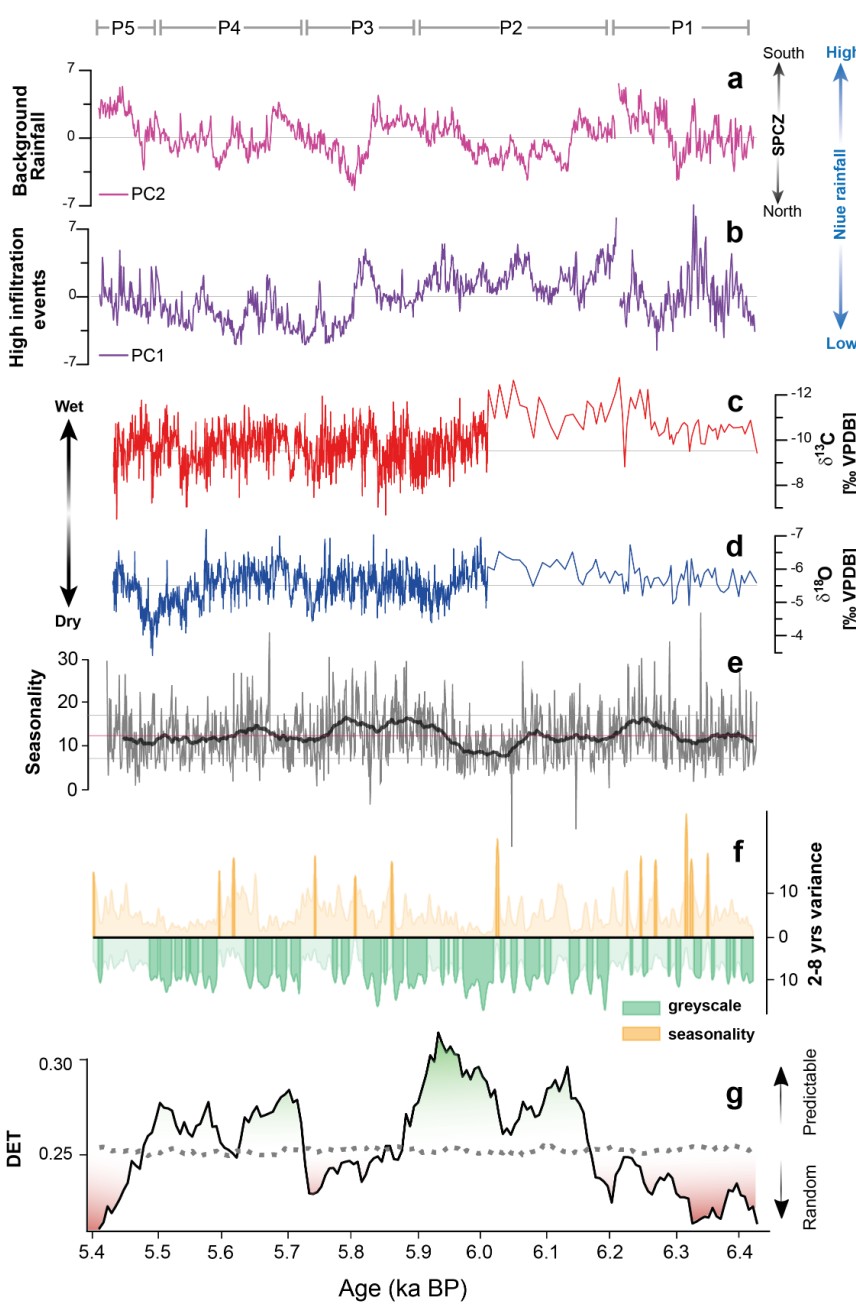

**Figure 7: a) Background rainfall (PC2). b) High infiltration events (PC1). c) δ¹³C record. d) δ¹⁸O record. e) Seasonality record extracted from the greyscale record. f) ENSO-scale variance computed as 2–8-year wavelet scale average (see fig. 7) from greyscale record (green) and seasonality index (yellow); darker-shaded peaks are above 95% significance. g) Seasonal predictability computed as DET from sliding recurrence plots; green shading indicates more predictable/regular seasonality and red shading indicates less predictable/random seasonality. See Sections 5.2 and 5.4 for calculation of (e)-(g). P1-P5 indicate different predictability phases.**
