# Peer review of "Mid-Holocene rainfall changes in the southwestern Pacific"

_Climate of the Past, 2021_

## Referee Comment (RC2)

Review of

**Mid-Holocene rainfall changes in the southwestern Pacific, v1**

by Nava-Fernandez et al.

**Recommendation**: *Accept with major revisions*

Summary The paper uses a rich data stream from an ultra-high (subannual) resolution record from Niue Island in the SPCZ region to infer changes in paleohydrology linked to ENSO and tropical cyclone activity. While the data are undoubtedly interesting, the interpretation is rather speculative, and conclusions overreach the level that can reasonably be made based on the presented evidence.

**1 Scientific Comments**

**1.1 Interpretation**

The paper's chief aim is to reconstruct paleohydrological conditions, which appear to be primarily influenced by ENSO and tropical cyclones. This mixed signal must be somehow deconvolved using proxies that cannot clearly separate those components.

Let me start with what I liked: the great diversity of observational techniques applied in the paper, resulting in information on stable isotopes of carbon and oxygen, optical measurements (grayscale) and trace element ratios. However, when trying to integrate these data streams, there is a bit of circular reasoning or cause for speculation. The biggest challenge is that no modern analog is presented, so no calibration is possible and none of the claims can be tested. Surely the authors would present a modern sample if they had one, but perhaps there are alternatives? For instance, the case for grayscale contrasts to measure the seasonality of precipitation is plausible, but would be a lot more convincing with a proof of concept at a similar location (even if not the same cave). Similarly, it is plausible that "Changes in stalagmite $\delta^{18}$O [...] provide information about the location of the SPCZ and/or the prevalence of tropical cyclones in the Central Pacific during the mid-Holocene", but this is only a qualitative claim, and where disagreements with other proxies (eg trace element PCs) appear, there is no guidance for how to resolve them. Also, such relationships are often frequency-dependent in the instrumental record, and there was no indication as to the timescales over which the $\delta^{18}$O interpretation might hold (interannual? decadal? longer?).

The arguments made to cluster trace elements and the hydrogeochemical processes leading to those groupings are also plausible, but would be a lot more convincing if buttressed by modern observations fro ma similar tropical catchment and/or a hydrochemical model.

Barring improvements in this direction, it will be difficult to take these records as more than a high-resolution mid-Holocene curiosity.

**1.2 Predictability**

Another major issue is with the concept of predictability. Recurrence quantification analysis is an interesting branch of timeseries analysis, and N. Marwan (a co-author) has a well-established history of developing such techniques and applying in the paleoclimate realm. However, the link between the "determinism" statistic and predictability is not made clearly. What sort of predictability is it anyway? It appears to be the ability to predict the occurrence of particular states based on past values of the series. However, this is quite distinct from most climate scientists' understanding of predictability, which involves a forecast model (even of the empirical kind) and a defined time horizon. What is the relationship between predictability as used in this paper and the seasonal predictability of Niue rainfall conditional on ENSO state, for instance? The lack of a precise definition of "predictability" works against the paper, making it seem like the authors want to give a veneer of rigor to an analysis that is ultimately fuzzy and qualitative. A revised manuscript should greatly clarify how this concept is to be understood by readers.

**1.3 Age modeling**

The paper goes above the primal age modeling impulse of most paleoclimatologists by using a probabilistic age model (COPRA) instead of simply connecting dots. However, the whole point of such a model is to sample the posterior distribution of the ages, not simply the median (which no real age-depth realization ever looks like). The authors do themselves (and their readers) a disservice by jettisoning this precious information. Note also that COPRA is an *ad hoc* method without a strong grounding in statistical theory. There are many better alternatives like BChron [*Haslett and Parnell*, 2008] or OxCal [*Bronk Ramsey*, 1995; *Ramsey*, 2008], both of which can be run easily in R through the GeoChronR package [*McKay et al.*, 2021], which also allows to take advantage of the whole posterior distribution of ages, not merely the median.

**2 Editorial Comments**

**P2, top** : "the influence that ENSO activity exerts on seasonality patterns" . Why not the other way around? Indeed, ENSO sits on top of the seasonal cycle, and often opposes it.

**P2, L54** "Diverse studies across the Pacific have reconstructed ENSO during the Holocene using climatic archives such as ..." also important to point out that these archives are not created equal when it comes to reconstructing ENSO. See for instance *Emile-Geay et al.* [2020].

**P3** "advanced statistical tools such as recurrence plots". That is a bit of an oversell. A recurrence plot is in fact a very simple object, and does not have its origins in statistics. Perhaps the authors can rephrase?

**P4** "El Niñoevents are associated with a northeastward displacement of the SPCZ and hence drier conditions in Niue". Some instrumental data need to be shown here, and more than just two events as done in Fig 1. A composite, or regression of rainfall on NINO3.4, would be helpful here.

**Section 4.1** "The age model of stalagmite C132 is confined". I think the authors mean "constrained", not "confined". Also, what is the uncertainty on the layer count? With this many laminae, it is bound to accrue errors, though presumably smaller ones that U/Th.

**Section 4.3** should explain the abrupt change in resolution ca 6000y BP. It is pointed out in the caption of Fig 3 but I did not find an explanation in the text - did I miss it?

**Section 5.1** "interpreted as common forcing" Not necessarily! PCA identifies patterns that explain the most variability, saying nothing about their intrinsic or extrinsic origin.

**L224** "The stable isotopes do not contribute to PC1 and PC2; instead, they present high loadings for PC4". This means that the isotopes provide orthogonal information, and this should be reflected in their interpretation.

**L230** "we performed a minor recalibration on the monthly-scale dating of the greyscale record". How is this dating done? It is not explained in 3.4.

**Section 5.3** is called spectral analysis, when in fact a wavelet analysis is shown. These are somewhat different – please fix.

**P11** "we interpret these elements as being mostly derived from marine aerosols" This is sensible but very qualitative. How can this be tied to changes in circulation/SPCZ location?

**Section 6.3** "We interpret these signals as indication for pronounced seasonality and low rainfall predictability due to significant influence of ENSO on atmospheric conditions over Niue." As detailed above, this is a highly ambiguous and controversial interpretation.

**P13, L397** "this stretch the seasonal cycle above normal" –> incorrect grammar. "It would be very unlikely that La Niña events result" –> incorrect grammar. L400: "confirmed" means that there was a prior hypothesis. Which one are you referring to?

**P14** The top paragraph is unintelligible. Please rephrase.

**P14L413** : "Niue record contribute with new evidence" –> incorrect grammar.

**P14L413** "throughout" –> through.

**Conclusion** "Investigated stalagmite from Niue Island" –> incorrect grammar. "two groups due to their source and mechanism of incorporation into calcite show high climatic sensitivity" –> incorrect grammar. "Importantly, tropical cyclone activity linked to El Niño links seasonal predictability with ENSO-band variability in overall background rainfall, with increased ENSO-band variability corresponding to lower seasonal predictability." –> a very bold conclusion unwarranted by the analysis.

**References**

Bronk Ramsey, C. (1995), Radiocarbon calibration and analysis of stratigraphy: The oxcal program, *Radiocarbon*, *37*(2), 425–430.

Emile-Geay, J., K. M. Cobb, J. E. Cole, M. Elliot, and F. Zhu (2020), *Past ENSO Variability*, chap. 5, pp. 87–118, American Geophysical Union (AGU), doi:10.1002/9781119548164.ch5.

Haslett, J., and A. Parnell (2008), A simple monotone process with application to radiocarbon-dated depth chronologies, *Journal of the Royal Statistical Society: Series C (Applied Statistics)*, *57*(4), 399–418, doi:10.1111/j.1467-9876.2008.00623.x.

McKay, N. P., J. Emile-Geay, and D. Khider (2021), geoChronR – an R package to model, analyze, and visualize age-uncertain data, *Geochronology*, *3*(1), 149–169, doi:10.5194/gchron-3-149-2021.

Ramsey, C. B. (2008), Deposition models for chronological records, *Quaternary Science Reviews*, *27*(1–2), 42 – 60, doi:10.1016/j.quascirev.2007.01.019, {INTegration} of Ice-core, Marine and Terrestrial records (INTIMATE): Refining the record of the Last Glacial-Interglacial Transition.